# Granzyme B inhibition reduces disease severity in autoimmune blistering diseases

Sho Hiroyasu [1,2,3], Matthew R. Zeglinski[1,2,3], Hongyan Zhao[1,2,3], Megan A. Pawluk [1,2,3], Christopher T. Turner [1,2,3], Anika Kasprick[4], Chiharu Tateishi[5], Wataru Nishie[6], Angela Burleigh[7], Peter A. Lennox[8], Nancy Van Laeken[8], Nick J. Carr[8], Frank Petersen[9], Richard I. Crawford[2,7], Hiroshi Shimizu[6], Daisuke Tsuruta[5], Ralf J. Ludwig [4] & David J. Granville [1,2,3 ✉]

Pemphigoid diseases refer to a group of severe autoimmune skin blistering diseases characterized by subepidermal blistering and loss of dermal-epidermal adhesion induced by autoantibody and immune cell infiltrate at the dermal-epidermal junction and upper dermis. Here, we explore the role of the immune cell-secreted serine protease, granzyme B, in pemphigoid disease pathogenesis using three independent murine models. In all models, granzyme B knockout or topical pharmacological inhibition significantly reduces total blistering area compared to controls. In vivo and in vitro studies show that granzyme B contributes to blistering by degrading key anchoring proteins in the dermal-epidermal junction that are necessary for dermal-epidermal adhesion. Further, granzyme B mediates IL-8/macrophage inflammatory protein-2 secretion, lesional neutrophil infiltration, and lesional neutrophil elastase activity. Clinically, granzyme B is elevated and abundant in human pemphigoid disease blister fluids and lesional skin. Collectively, granzyme B is a potential therapeutic target in pemphigoid diseases.

[1] International Collaboration On Repair Discoveries (ICORD) Centre, Vancouver Coastal Health Research Institute (VCHRI), Vancouver, BC, Canada. [2] Department of Pathology and Laboratory Medicine, University of British Columbia (UBC), Vancouver, BC, Canada. [3] BC Professional Firefighters' Burn and Wound Healing Research Laboratory, VCHRI, Vancouver, BC, Canada. [4] Lübeck Institute of Experimental Dermatology, University of Lübeck, Lübeck, Germany. [5] Department of Dermatology, Osaka City University Graduate School of Medicine, Osaka, Japan. [6] Department of Dermatology, Hokkaido University Graduate School of Medicine, Sapporo, Japan. [7] Department of Dermatology and Skin Science, UBC, Vancouver, BC, Canada. [8] Department of Surgery, UBC, Vancouver, BC, Canada. [9] Priority Area Asthma and Allergy, Members of the German Center for Lung Research, Research Center Borstel, Borstel, Germany. ✉email: David.Granville@hli.ubc.ca

Pemphigoid diseases (PDs), including epidermolysis bullosa acquisita (EBA) and bullous pemphigoid (BP), are auto-immune subepidermal blistering diseases caused by the development of autoantibodies against proteins at the dermal–epidermal junction[1]. Autoantibody accumulation at the dermal–epidermal junction induces the generation and release of cytokines and complement, followed by inflammatory cell infiltration into the upper dermis (reviewed in ref. [2]). In the upper dermis, inflammatory cells release proteases that cleave anchoring proteins at the dermal–epidermal junction to separate the epidermis from the dermis, resulting in erythema, blistering, and erosions. Presently, the incidence of BP, the most common PD, is escalating as a result of improved diagnostic tools and increasing prevalence of risk factors such as aging, neurological conditions, and exposure to triggering drugs (reviewed in ref. [3]). The current standard of treatment for PDs is topical or oral corticosteroid administration, which non-specifically targets the inflammatory response and is often associated with severe and potentially fatal adverse events[4]. Therefore, effective, targeted treatment options are an unmet clinical need.

Proteases underlie many pathological mechanisms in PDs and were previously considered promising therapeutic targets for this group of diseases (reviewed in ref. [5]). Specifically, it was proposed that neutrophil elastase induces subepidermal blistering in PDs through proteolytic degradation of hemidesmosomal proteins, which form an anchoring complex between the epidermis and dermis[6]. Other proteases such as matrix metalloprotease-9 (MMP-9) and plasmin/plasminogen/plasminogen activators (PAs) have also been proposed to contribute to PD pathology by regulating the proteolytic activity of neutrophil elastase[7]. Nevertheless, treatment options targeting these proteases in PDs have not progressed to clinical trials, likely due to severe adverse effects and/or limited therapeutic efficacy[5,8,9].

Granzyme B (GzmB) is an immune and non-immune cell-secreted serine protease that is abundant in the lesional areas of PD patient skin biopsies[10,11]. Unlike other proteases, the pathological role of GzmB in PDs has not been explored, and this may be attributed to the long-held misperception that GzmB functions exclusively as a cytotoxic, pro-apoptotic protease (reviewed in ref. [12]). In this traditional dogma, GzmB was considered to be exclusively released from the granules of cytotoxic T and natural killer cells and internalized into target cells through perforin-mediated pores to initiate apoptosis. However, it is now recognized that other immune and non-immune cell types also secrete GzmB, independent of perforin, to mediate non-cytotoxic, extracellular functions[13–16]. Unlike other proteases such as neutrophil elastase and MMPs, GzmB has no known endogenous extracellular inhibitor[17]; therefore GzmB accumulates and retains its proteolytic activity in the extracellular space. While minimally expressed or absent in healthy, non-inflamed tissues, unimpeded GzmB-mediated proteolytic activity in the extracellular space can degrade extracellular proteins and regulate immune responses during inflammation, thereby contributing to pathogenesis[18,19]. Therefore, we hypothesized that extracellular GzmB exerts a pathological role in PDs through the degradation of hemidesmosomal proteins and promotion of inflammation, and is a therapeutic target for PDs.

In the present study, the function of GzmB was assessed in passive IgG transfer murine models of EBA and BP. Our data indicate that genetic deficiency or topical inhibition of GzmB in PDs reduces disease severity, prevents hemidesmosomal protein loss, impedes neutrophil infiltration, and impairs lesional neutrophil elastase activity.

## Results

### GzmB deficiency reduces disease severity in an inflammatory EBA murine model.
To investigate the role of GzmB in PD pathogenesis, an inflammatory EBA murine model was induced through intraperitoneal transfer of pathogenic anti-type VII collagen (COL7) IgG into both wild-type (WT) and GzmB−/− mice based on an established method[20]. While both WT and GzmB−/− mice with EBA exhibited skin lesions mainly at the ears, head, face, neck, and legs at day 12, GzmB−/− mice with EBA exhibited a 45% reduction in lesional area compared to WT mice with EBA (Fig. 1a, b). This observation was consistent with histological analyses of the ears at day 12 whereby GzmB−/− mice with EBA showed a 55% reduction in the mean histological blistering score compared to their WT counterparts (Fig. 1a, c). Histological analysis of ears from both groups showed comparable degrees of dermal inflammatory cell infiltration (Fig. 1a). Sequential staining studies revealed that toluidine blue O (TBO)-positive cells, mast cells and basophils[21], were major contributors of GzmB production in the upper dermis of WT mice with EBA but not GzmB−/− mice with EBA, normal IgG-injected WT mice, or normal IgG-injected GzmB−/− mice (Fig. 1d and Supplementary Fig. 1a). Double immunostaining of GzmB and a mouse basophil-specific marker, mouse mast cell protease-8 (mMCP-8)[22], in WT mice with EBA, showed a subset of GzmB-positive cells was also mMCP-8 positive, which supported our findings with TBO staining that not only mast cells but also basophils were major sources of GzmB (Fig. 1e). Double immunostaining of GzmB and mMCP-8 in mast cell-deficient mice (diphtheria toxin (DT)-treated Mcpt5-Cre iDTR mice) with EBA further confirmed that basophils were a source of GzmB (Supplementary Fig. 1b).

### GzmB mediates loss of hemidesmosomal proteins, type XVII collagen (COL17) and α6 integrin, in an EBA murine model.
We next aimed to elucidate the mechanisms underlying GzmB-mediated PD pathology. GzmB was previously reported to be elevated in human PDs while its substrates, COL17 and α6 integrin, were reduced at the lesions[10]. However, a causal link between GzmB elevation and hemidesmosomal protein loss in PDs was not determined. Here, the involvement of GzmB in the loss of hemidesmosomal proteins was investigated in the EBA murine model described above. Murine ears were assessed for COL17 and α6 integrin by immunohistochemistry (IHC) at day 12 (Fig. 2a). Epidermal COL17 and α6 integrin staining intensity decreased by ~60% in WT mice with EBA compared to the normal IgG-injected WT mice and GzmB−/− mice with EBA. Intensities of protein staining were quantified only in epidermis not separated from the dermis in order to exclude the possible contribution of anoikis-altered protein levels. Levels of COL17 protein from the ear extracts of the mice at day 12 were further analyzed by immunoblotting (Fig. 2b). While WT mice with EBA exhibited decreased COL17 protein levels at 50% of the normal IgG-injected WT mice, GzmB−/− mice with EBA showed comparable levels of COL17 protein to normal IgG-injected GzmB−/− and normal IgG-injected WT mice. Protein levels of α6 integrin in the ear extracts were not assessed by immunoblotting as α6 integrin is expressed by many cell types, including neutrophils, that do not directly influence dermal–epidermal attachment[23].

To investigate the direct effects of GzmB on hemidesmosomal proteins, COL17 and α6 integrin proteins were analyzed in normal primary human epidermal keratinocytes (pHEKs) with GzmB treatment (Fig. 2c). After 6-h incubations with up to 50 nM GzmB, COL17 was cleaved in a dose-dependent manner as detected by an antibody against the NC16A domain of COL17. When GzmB was pre-treated with a potent GzmB-specific inhibitor, VTI-1002 (ref. [24]), for 30 min before incubation with the pHEKs, degradation of COL17 was inhibited. Direct COL17

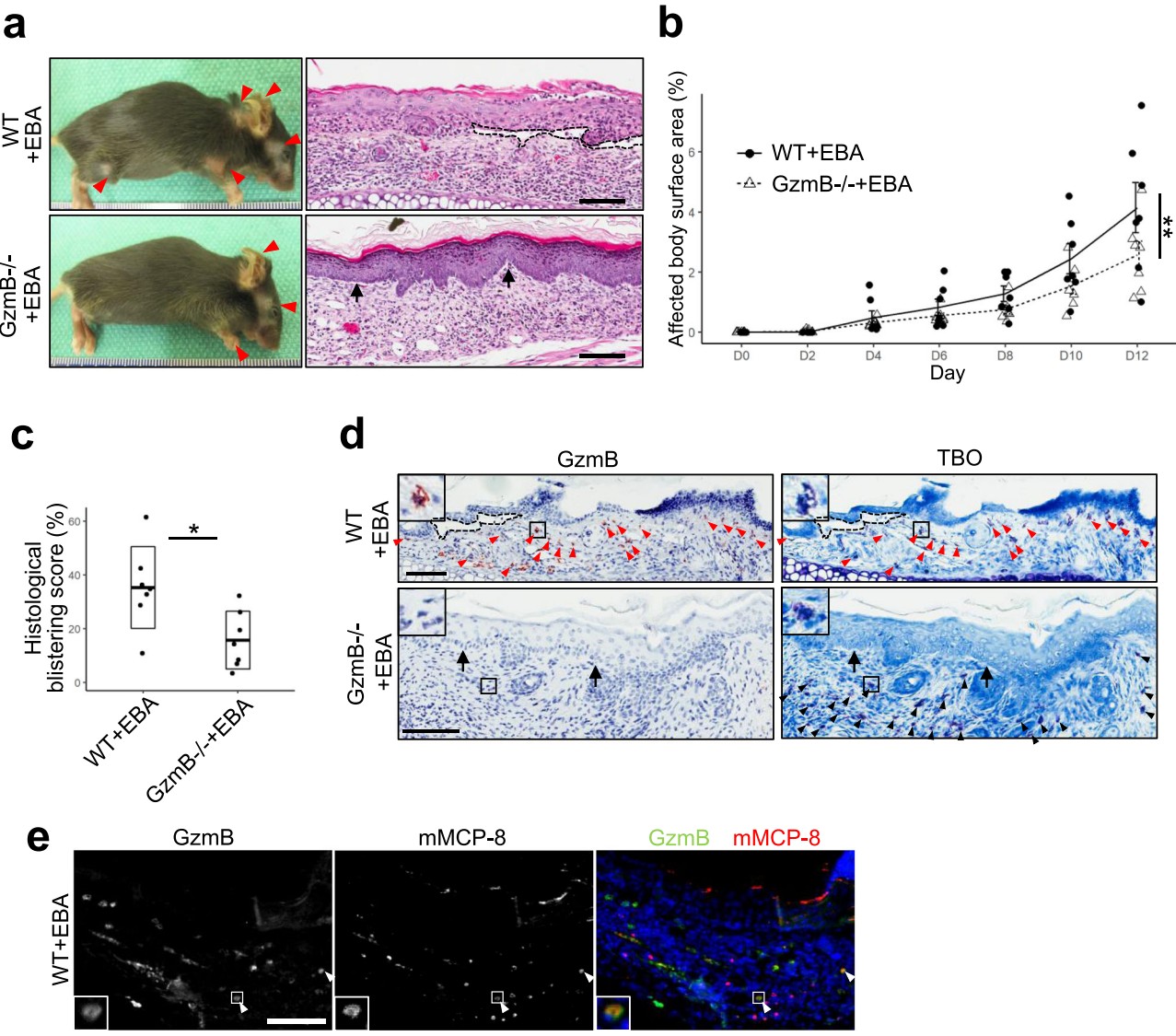

**Fig. 1 Granzyme B (GzmB)−/− murine model of inflammatory epidermolysis bullosa acquisita (EBA) exhibits reduced blistering. a** Representative clinical and H&E staining images of wild-type (WT) and GzmB−/− mice with EBA at day 12 induced by intraperitoneal injections of pathogenic anti-type VII collagen (COL7)-IgG, labeled as WT+EBA and GzmB−/−+EBA in the figures, respectively. Red arrowheads indicate clinical lesions. Dotted line demarcates major dermal–epidermal separation and black arrows indicate minor dermal–epidermal separation. Scale bar, 100 μm. **b** Affected body surface area defined as (total affected body surface area)/(total body surface area) × 100 was quantified every 2 days for 12 days. Dot plots indicate all individual scores and mean ± standard error. N = 7 for each strain. **P < 0.01 (two-way ANOVA (time and strain as two variables)). **c** Histological blistering scores defined as (combined total length of all blistered regions)/(combined total length of all dermal–epidermal junction examined) were quantified from H&E staining images at day 12. Dot plots indicate all individual scores and box plots indicate mean ± standard deviation. N = 7 for each strain. *P < 0.05 (two-sided Student's t-test). **d** Representative sequential GzmB immunohistochemistry (IHC) and toluidine blue O (TBO) staining images of ears from WT and GzmB−/− mice with EBA at day 12. TBO stains mast cells and basophils dark blue to purple. Red arrowheads indicate cells stained with both GzmB and TBO. Black arrowheads indicate cells stained with TBO but not GzmB. Dotted line demarcates major dermal–epidermal separation and black arrows indicate minor dermal–epidermal separation. Scale bar, 100 μm. **e** Representative double staining images of ears from WT mice with EBA at day 12 with antibodies against GzmB and mouse basophil marker mouse mast cell protease-8 (mMCP-8). Third column shows overlays of two stains. White arrowheads indicate cells stained with both GzmB and mMCP-8. Scale bar, 100 μm. Images in **a**, **d**, and **e** are representative of two (**a**) or three (**d**, **e**) independent experiments.

proteolytic cleavage by GzmB was validated in a cell-free system using recombinant human COL17 and GzmB (Supplementary Fig. 1c). In contrast to COL17, protein levels of α6 integrin in pHEKs did not show significant changes with up to 50 nM GzmB (Fig. 2c). This suggests that the GzmB-induced COL17 and α6 integrin protein loss in our murine model was not only a result of direct proteolytic degradation but also other mechanisms, such as

inflammatory regulation. Consistent with decreased COL17 protein levels, pHEKs exhibited reduced adherence to the culture plate after 6-h treatments with GzmB in a trypsinization assay, with 65% of cells adherent when normalized to 0 nM GzmB-treated cells (Fig. 2d). GzmB-induced loss of adherence in pHEKs was partially offset with VTI-1002 pretreatment. MTT (3-(4,5-dimethylthiazol-2-yl)-2,5-diphenyltetrazolium bromide) assay

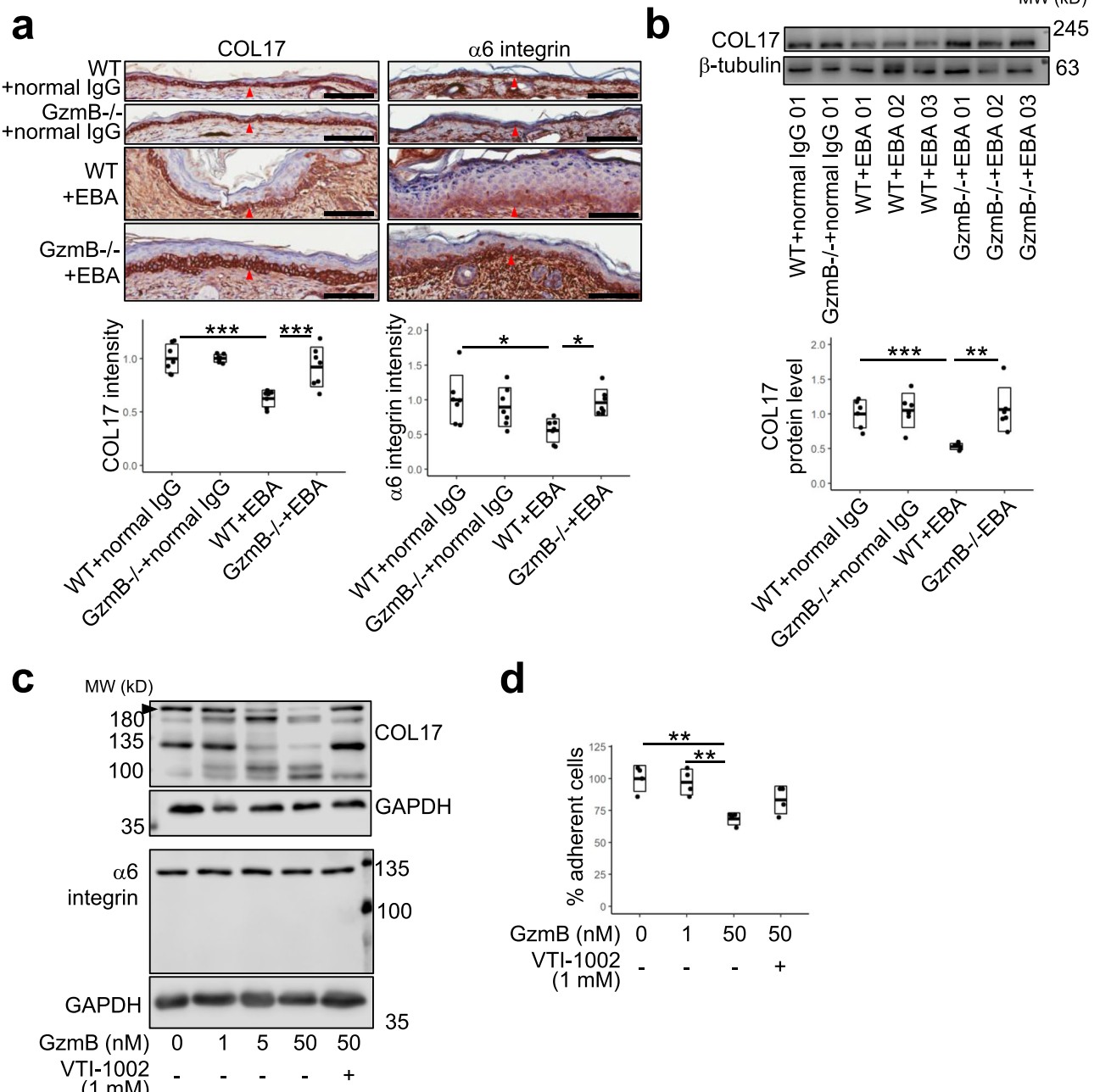

**Fig. 2 Granzyme B (GzmB) reduces hemidesmosomal proteins, type XVII collagen (COL17) and α6 integrin, in an epidermolysis bullosa acquisita (EBA) murine model in vivo and in cultured primary human epidermal keratinocyte (pHEK) in vitro. a** Representative COL17 and α6 integrin immunohistochemistry (IHC) images of ears from both wild-type (WT) and GzmB−/− mice at day 12 with EBA or with injections of normal rabbit IgG. Red arrowheads indicate positively stained areas at dermal–epidermal junction. Scale bar, 100 μm. Dot plots indicate all individual scores and box plots indicate mean ± standard deviation of the intensity levels of hemidesmosomal proteins in the epidermis defined as (total intensity of hemidesmosomal protein in the epidermis)/(total area of examined epidermis), which were quantified from IHC images and presented as relative to the mean score of WT with normal IgG injections. $N = 7$ for each group. **b** Representative immunoblotting images using an antibody against COL17 for ear extracts from both WT and GzmB−/− mice at day 12 with EBA or with normal IgG injections. Same amounts of the same extracts were loaded on a separate gel and reactivity to β-tubulin antibody was used as sample processing control. Dot plots indicate all individual scores and box plots indicate mean ± standard deviation of COL17 protein levels normalized to the β-tubulin level, which were quantified from immunoblots and presented relative to the mean score of WT with normal IgG injections. $N = 6$ for each group. **c** Representative immunoblotting images using antibodies against COL17 NC16A domain or α6 integrin with cell lysates of pHEKs, which were incubated with the GzmB inhibitor, VTI-1002- or vehicle-pretreated GzmB for 6 h. Black arrowhead indicates the bands of full-length COL17. Same amounts of the same extracts were loaded on the same or separate gel and glyceraldehyde 3-phosphate dehydrogenase (GAPDH) served as sample processing control. Molecular weight (MW) is indicated in the sides. Images are representative of three independent experiments. **d** pHEKs were incubated with VTI-1002- or vehicle-pretreated GzmB for 6 h followed by trypsinization for 7 min at 37 °C and the attached cells were counted. Cell numbers were normalized to the average number of attached cells incubated without GzmB or VTI-1002. Dot plots indicate all individual scores and box plots indicate mean ± standard deviation. $N = 4$ for each group. In all plots, *$P < 0.05$, **$P < 0.01$, ***$P < 0.001$ (one-way ANOVA followed by Tukey's multiple pairwise-comparisons).

results confirmed that GzmB did not affect pHEK cell viability (Supplementary Fig. 1d).

**GzmB deficiency hinders neutrophil infiltration through impeded secretion of chemoattractant macrophage inflammatory protein-2/IL-8.** Next, the effects of GzmB on the inflammatory response in the EBA murine model were assessed. As the effector phase of PDs is initiated by antibody deposition followed by complement accumulation at the dermal–epidermal junction, IgG and complement C3 were assessed in WT and GzmB−/− murine ears with EBA (Fig. 3a). Linear deposition patterns of IgG and C3 at the dermal–epidermal junction were observed in both mouse strains with EBA. No obvious difference in the patterns or intensities was observed between the groups, suggesting that GzmB does not affect IgG or complement deposition.

As neutrophils are key mediators of PD blistering after complement activation at the dermal–epidermal junction through neutrophil elastase-dependent hemidesmosomal protein degradation[6], neutrophil infiltration was evaluated using neutrophil elastase immunostaining on WT and GzmB−/− ears with EBA (Fig. 3b). While the total number of infiltrated cells in the dermis was comparable between the two strains, a 40% decrease in neutrophils was observed in GzmB−/− mice with EBA compared to WT mice with EBA. Levels of the major neutrophil chemoattractant, macrophage inflammatory protein-2 (MIP-2; mouse homolog of IL-8), were significantly elevated in the extracts of lesional skin of WT mice with EBA compared to the normal IgG-injected mice (0 pg/mg), while levels of MIP-2 in the skin of GzmB−/− mice with EBA (1000 pg/mg) were decreased when compared to the WT mice with EBA (1900 pg/mg) (Fig. 3c). To further test if GzmB positively regulated MIP-2 production and to ascertain keratinocytes as a cell source of MIP-2, MIP-2 was quantified in the supernatants of primary mouse epidermal keratinocytes after GzmB stimulation for 16 h. MIP-2 levels in the supernatant were increased in a GzmB-dose dependent manner while GzmB inhibitor VTI-1002 attenuated GzmB-induced MIP-2 secretion (Fig. 3d). Levels of IL-8, MIP-2 human homolog, in pHEK supernatant were also elevated in a GzmB-dose dependent manner (Supplementary Fig. 1g).

Consistent with the observed reduced neutrophil infiltration, neutrophil elastase activity in the lesional skin was decreased in GzmB−/− mice with EBA as detected by a neutrophil elastase-specific fluorogenic substrate (Fig. 3e). In contrast, the activities of plasmin and MMP-1/MMP-9 were comparable between WT and GzmB−/− mice with EBA. As GzmB did not directly degrade α6 integrin in pHEKs (Fig. 2c), we postulated that GzmB modulated α6 integrin loss in WT EBA mice by promoting neutrophil infiltration and neutrophil elastase activity. To support this model, 50 nM of neutrophil elastase degraded α6 integrin in pHEKs after a 6-h incubation (Supplementary Fig. 1e), and no change in pHEK viability was detected (Supplementary Fig. 1f).

**Topical application of GzmB inhibitor VTI-1002 reduces disease severity in a local antibody-transfer model of inflammatory EBA.** Topical application of VTI-1002 was previously demonstrated to be safe, efficacious, and retained in murine skin for up to 24 h in a murine model of burn wound healing[24]. As we elucidated that GzmB pathogenesis in PDs was mediated through COL17 degradation and MIP-2/IL-8 elevation at the lesional skin but not through IgG recruitment, the simpler and less-invasive local antibody-transfer model of inflammatory EBA was chosen to assess the efficacy of topical VTI-1002. The model was induced with subcutaneous injections of pathogenic anti-COL7 IgG into the ear tips[20]. Sequential staining of TBO and GzmB in addition to double staining of mMCP-8 and GzmB on the ears of the local

EBA model identified increased GzmB production in the mast cells and basophils of the dermis, which was consistent with findings from the systemic antibody-transfer EBA model (Supplementary Fig. 2a, b). Vehicle control or VTI-1002 gel was applied daily onto the ears beginning 1 day prior to IgG injections. At day 3, VTI-1002 gel-treated EBA ears showed 50% reduction in lesional areas compared to EBA ears treated with vehicle gel (Fig. 4a, b). Ear swelling estimated by ear thickness measurements was also reduced in the EBA ears with VTI-1002 gel treatment compared to the vehicle gel-treated EBA ears from a mean of 0.35 mm to 0.29 mm, suggesting decreased inflammation (Fig. 4c). Histological analysis corresponded to these observations as the areas of subepidermal separation were 55% decreased in VTI-1002 gel-treated EBA ears compared to the vehicle gel-treated EBA ears (Fig. 4a, d). Protein levels of COL17 and α6 integrin in the epidermis were increased in VTI-1002 gel-treated EBA ears to 120% and 160%, respectively, compared to the vehicle gel-treated EBA ears, as determined by IHC analysis (Fig. 4e). IHC analysis for neutrophil elastase with nuclear staining indicated that VTI-1002 gel treatment decreased neutrophil infiltration by 40% but not total cell infiltration (Fig. 4f).

**Topical application of VTI-1002 reduces disease severity in a BP murine model.** BP is the most common PD and is caused by autoantibodies against COL17. To validate the therapeutic efficacy of VTI-1002 gel in BP, a pathogenic human BP-IgG transfer model of BP was induced into neonatal humanized-COL17 mice[25]. Consistent with the EBA models, BP mice showed greater numbers of TBO-positive cells expressing GzmB in the dermis (Supplementary Fig. 2c). Surprisingly, we did not detect any GzmB-expressing basophils with double staining of mMCP-8 and GzmB (Supplementary Fig. 2d). In contrast to the systemic and local EBA murine models, marginal levels of basophils were detected in BP neonatal mice with mMCP-8 staining (Supplementary Fig. 2e). These findings suggested that mast cells but not basophils were a significant source of GzmB in this neonatal BP model.

BP mice were treated with vehicle or VTI-1002 gel daily, starting 1 day prior to IgG injections. VTI-1002 gel-treated mice exhibited markedly reduced skin fragility compared to the vehicle gel-treated mice (Fig. 5a and Table 1). Nikolsky's sign (skin detachment caused by gentle rubbing) was observed in nine out of nine mice treated with vehicle gel alone, with seven mice exhibiting bilateral skin detachment. Conversely, Nikolsky's sign was observed in only three of nine mice treated with VTI-1002 gel, with one mouse showing bilateral skin detachment. This result was consistent with the histological blistering scores measuring dermal–epidermal separation in VTI-1002 gel-treated (5%) and vehicle gel-treated BP mice (25%) (Fig. 5a, b). Protein levels of COL17 and α6 integrin in the epidermis were increased in VTI-1002 gel-treated BP mice by 140% compared to vehicle gel-treated BP mice (Fig. 5c) while neutrophil elastase and nuclear staining indicated that neutrophil infiltration in VTI-1002 gel-treated BP mice was reduced by 30% compared to vehicle gel-treated mice (Fig. 5d).

**GzmB levels are elevated in primary human BP blister fluids and lesional skin.** GzmB was investigated in primary control sera, BP patient sera, BP blister fluids, and BP lesional skins. Protein levels of GzmB in BP blister fluids ranged from approximately 200 to 750 pg/ml, while it was negligible in both BP patient sera and healthy control sera (Fig. 6a). GzmB-positive cells were localized at the dermis of the primary BP patient lesional and perilesional skin and stained positive for TBO, indicating mast cells or basophils (Fig. 6b).

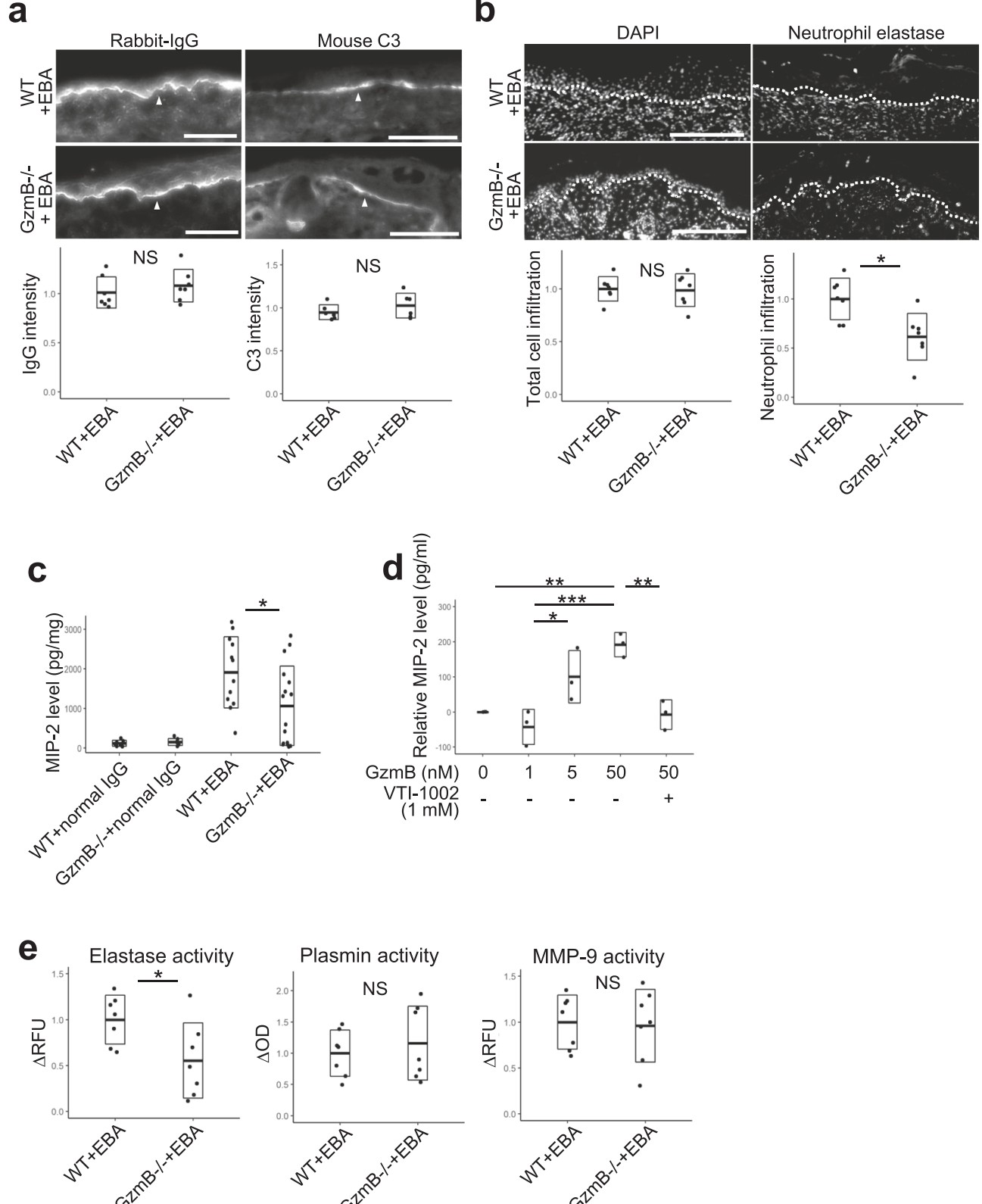

Double immunostaining of GzmB and human basophil specific marker, pro-major basic protein 1 (ProMBP1)[26], identified that varying numbers of GzmB-expressing basophils were present in the majority of patient lesional and perilesional skins (Fig. 6c, d). Based on a previous study indicating human basophils are activated through the IgE receptor[27], we assessed if levels of

deposited IgE correlate to the presence of GzmB-expressing basophils in patient skin. IHC analysis of IgE indicated that IgE was deposited at the infiltrated cells of the lesional and perilesional dermis and the deposition positively correlated to the number of GzmB-positive basophils (Fig. 6c, d). Taken together, we posit that in PDs, the extracellular GzmB produced in mast

**Fig. 3 Granzyme B (GzmB) promotes neutrophil infiltration, increased macrophage inflammatory protein-2 (MIP-2)/IL-8 levels, and elevated elastase activity. a** Representative direct immunofluorescence (DIF) images of rabbit IgG or mouse complement C3 of ears from both wild-type (WT) and GzmB−/− mice with epidermolysis bullosa acquisita (EBA) at day 12. Scale bar, 100 μm. Dot plots indicate all individual scores and box plots indicate mean ± standard deviation of the relative intensities of IgG or C3 at the dermal–epidermal junctions, which were quantified from DIF images and presented relative to the mean scores in WT with EBA. N = 7 for each group. **b** Representative immunohistochemistry (IHC) images of ears from both WT and GzmB−/− mice with EBA at day 12 stained with neutrophil elastase antibody and 4′,6-diamidino-2-phenylindole (DAPI). White dotted lines indicate the dermal–epidermal junction. Scale bar, 100 μm. Left and right dot plots indicate all individual scores and box plots indicate mean ± standard deviation of relative total cell infiltration score ((total DAPI-positive area in upper dermis)/(total area of upper dermis)) and relative neutrophil infiltrating score ((total neutrophil elastase-positive area in upper dermis)/(total DAPI-positive area in upper dermis)), respectively. Both scores are presented relative to the mean scores in WT with EBA. N = 7 for each group. **c** MIP-2 levels in the lesional skins of WT and GzmB−/− mice with EBA at day 12 were assessed with enzyme-linked immunosorbent assays (ELISA). Dot plot indicates all scores and box plots indicate mean ± standard deviation. N = 12 for each group with EBA and N = 7 for each control group. **d** Primary mouse keratinocytes were incubated with VTI-1002 or vehicle-pretreated GzmB for 16 h and MIP-2 levels in the supernatants were assessed with ELISA. Dot plot indicates MIP-2 levels of the primary mouse keratinocyte supernatant without GzmB stimulation subtracted from measured MIP-2 levels, and box plots indicate the means ± standard deviation. N = 3 for each group. In all plots in **c** and **d**, *P < 0.05, **P < 0.01, ***P < 0.001 (one-way ANOVA with Tukey's multiple pairwise-comparisons). **e** Protease activities of elastase, plasmin, and matrix metalloprotease-9 (MMP-9) in the lesional skins of WT and GzmB−/− mice with EBA at day 12 were assessed. Δ relative fluorescent unit (RFU) and Δ optical density (OD) were calculated by subtracting scores at time 0 from scores after 30-min incubation. Dot plots indicate all scores and box plots indicate mean ± standard deviation. N = 7 for each group. In all plots in **a**, **b**, and **e**, NS = not significant, *P < 0.05 (two-sided Student's t-test).

cells and/or basophils leads to degradation of hemidesmosomal proteins and subsequent subepidermal blistering (Fig. 6e). This phenotype is further exacerbated by the induction of IL-8/MIP-2 chemoattractant by GzmB from keratinocytes, resulting in neutrophil infiltration and additional proteolytic cleavage of hemidesmosomal protein by neutrophil elastase.

## Discussion

In PD pathology, proteases such as neutrophil elastase, matrix metalloproteinases (MMPs), and plasmin/plasminogen/PAs have been investigated as mediators of proteolytic degradation of hemidesmosomal proteins and augmentation of other proteases[7,28–30] (reviewed in ref. [5]). However, their utility as therapeutic targets for inflammatory diseases has been limited possibly due to limited therapeutic efficacy and/or induction of adverse effects[5,8,9]. GzmB, a serine protease, is highly elevated in lesional skin of PD patients[10,11]. The present study is the first to report a key causal role of GzmB in PD pathogenesis, demonstrating that GzmB contributes to increased disease severity in murine PD models through hemidesmosomal protein degradation and MIP-2 induction (Fig. 6d). In addition, the therapeutic efficacy of a topical GzmB inhibitor, VTI-1002 gel, was demonstrated using two PD murine models.

In the current study, GzmB cleaved COL17 in vivo and in vitro, reducing epidermal attachment to the extracellular matrix, while GzmB (50 nM) failed to degrade α6 integrin expressed in pHEKs. Previously, we have shown that GzmB (200 nM) cleaves the recombinant extracellular domain of α6 integrin in an in vitro, cell-free assay[10]. In a cell culture model, it is likely that structural protein conformation and/or other hemidesmosomal proteins could be blocking access to the cleavage sites of α6 integrin to prevent proteolytic cleavage. Nonetheless, in the murine model, GzmB decreased protein levels of both COL17 and α6 integrin in the systemic antibody-transfer EBA murine model. As proteolytic degradation of α6 integrin by other proteases has been reported[31], we hypothesized that GzmB augments inflammation to increase the activity of other proteases to degrade α6 integrin. To test this, we evaluated the activity of a number of proteases in GzmB−/− mice with EBA and identified a reduction in neutrophil elastase activity compared to WT mice with EBA. Furthermore, we revealed that neutrophil elastase degraded α6 integrin expressed in pHEKs, collectively suggesting that GzmB contributes to the loss of α6 integrin through increasing neutrophil elastase activity in EBA.

Consistent with GzmB-dependent elevation of neutrophil elastase activity in EBA, we identified GzmB-dependent infiltration of neutrophils, a predominant source of neutrophil elastase[32], in EBA mice and GzmB-dependent elevation of strong neutrophil chemoattractant MIP-2, a mouse homolog of human IL-8[33]. IL-8 has been implicated as a key mediator of PD blistering. This is evidenced by a positive correlation between concentrations of IL-8 in BP blister fluid and disease severity[34], as well as findings that intradermal IL-8 injections restore BP blistering in disease-resistant mice with deficiency of complement or with Kit/Scf-dependent deficiency of mast cells[35–37]. In our current study, GzmB stimulation increased IL-8 and MIP-2 secretion by pHEKs and primary mouse epidermal keratinocytes, respectively. As GzmB induces COL17 loss in keratinocytes and keratinocyte secretion of IL-8 correlates to COL17 loss[38], GzmB may induce IL-8 secretion from keratinocytes in response to COL17 degradation. While this is the first report describing GzmB-mediated induction of neutrophil elastase activity to enhance hemidesmosomal protein loss in PDs, this does not exclude potential interactions between GzmB and other proteases in PDs. Indeed, a recent work revealed that GzmB activates caspase 3 in secretory lysosomes of mast cells, a mechanism which possibly contributes to enhanced caspase 3-dependent proteolytic cleavage in the extracellular space[39]. Since recent studies combining genomics, proteomics, and bioinformatics are beginning to elucidate the direct and indirect mutual influence of proteases[40], our understanding of the complex interactions between GzmB and other proteases in PDs will be further refined.

BP patient samples and the murine models of PDs indicate that GzmB is produced in PDs by TBO-positive mast cells and/or basophils. This is consistent with previous reports demonstrating that activated mast cells and basophils can express GzmB but do not produce perforin[13,14,41]. Unexpectedly, subtle differences in the infiltrated immune cell sources of GzmB were identified between the described murine models and patient specimens in our current study. Specifically, although both mast cell and basophil infiltrations were observed in the adult EBA murine models and human BP patients, only mast cell infiltration was observed in the neonatal BP murine model. In addition, while both mast cells and basophils exhibited GzmB immunopositivity in the adult EBA models, varying numbers of GzmB-expressing basophils were detected in the human BP samples.

While discrepancies in the pathology between adult EBA and neonatal BP models have previously been identified, including BP-IgG-induced COL17 internalization in BP[42–44], underlying

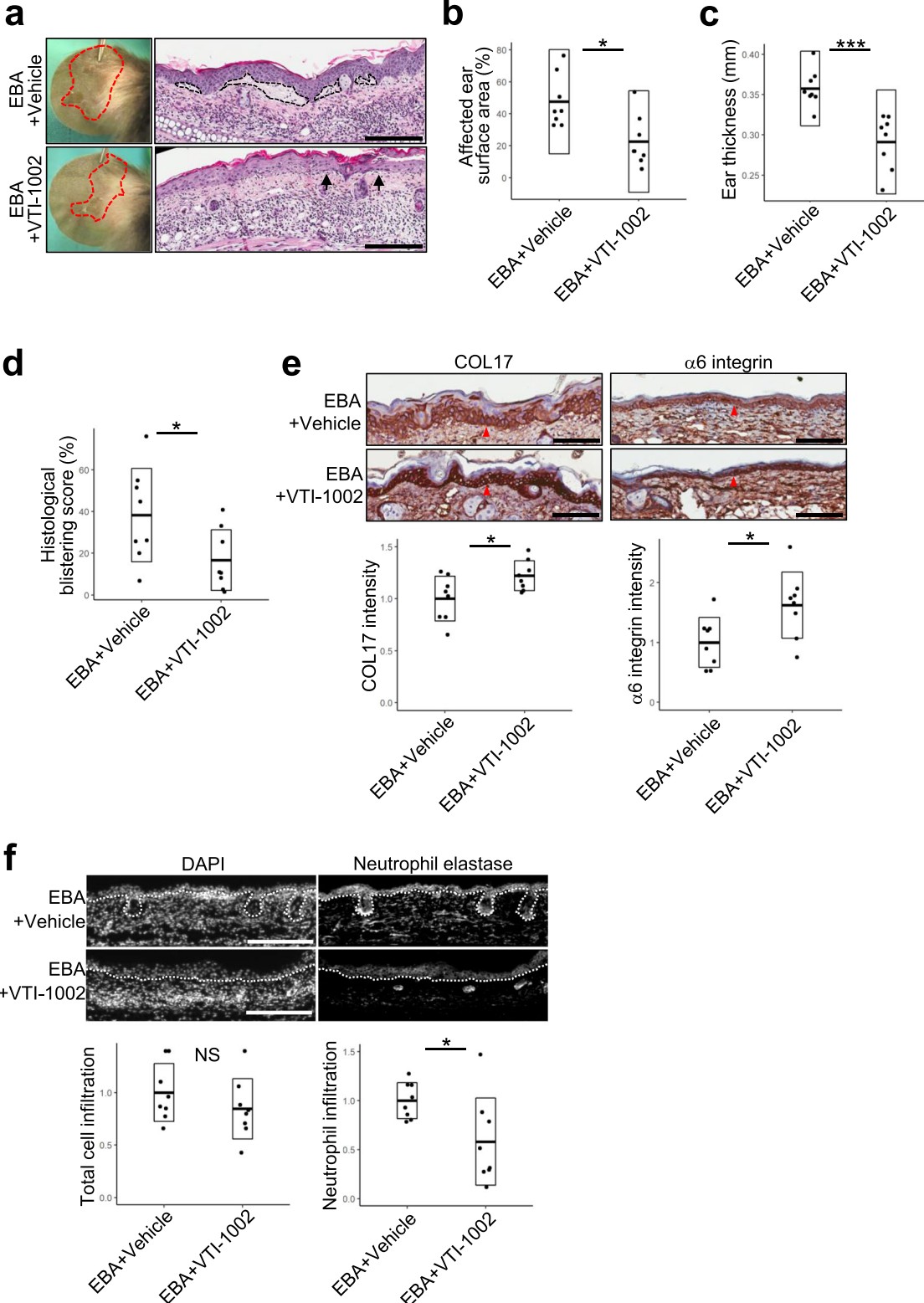

mechanisms to explain the distinct phenotypes with mast cell deficiency have been elusive until now. In particular, while mast cell deficiency generated no blisters in an anti-mouse COL17 IgG transfer neonatal BP model[37], mast cell-deficient mice exhibited consistent blistering phenotypes in an anti-mouse COL7 IgG transfer adult EBA model[45]. Our present findings demonstrate that distinct but overlapping immune cell populations infiltrate the lesions in the described PD models. This is the first report to

suggest that mast cell deficiency may not be sufficient to deplete pathogenic GzmB (and possibly other proteases produced by both mast cells and basophils) in the adult EBA model due to basophil infiltration, while mast cell deficiency may sufficiently deplete GzmB (and possibly other proteases) in the neonatal BP model.

As the pathological function of basophils in human PDs has not been explored except a histological study by others detecting its infiltration in the BP lesions[46], the current study proposes a

**Fig. 4 Topical application of VTI-1002 gel lessens disease severity in a local-passive transfer epidermolysis bullosa acquisita (EBA) murine model.**
**a** Representative clinical and H&E staining images of both vehicle- and VTI-1002 gel-treated local EBA mouse ears at day 3 induced by subcutaneous anti-type VII collagen (COL7)-IgG injections, labeled as EBA+Vehicle and EBA+VTI-1002 in the figure, respectively. Red dotted lines delineate the lesional areas. Dotted lines demarcate major dermal–epidermal separation and black arrows indicate minor dermal–epidermal separation. Scale bar, 100 μm.
**b** Affected ear surface area defined as (total affected ear surface area)/(total ear surface area) × 100 and **c** ear thickness were quantified at day 3. Dot plots indicate all individual scores and box plots indicate mean ± standard deviation. $N = 8$ for each group. **d** Histological blistering scores were quantified from H&E staining images. Dot plot indicates all scores and box plots indicate mean ± standard deviation. $N = 8$ for each group. **e** Representative type XVII collagen (COL17) and α6 integrin immunohistochemistry (IHC) images of both vehicle and VTI-1002 gel-treated local EBA mouse ears at day 3. Red arrowheads indicate positively stained areas at dermal–epidermal junction. Scale bar, 100 μm. Dot plot indicates all individual scores and mean ± standard deviation of the intensity levels of COL17 and α6 integrin in the epidermis, which were quantified from IHC images and presented as relative to the mean score of vehicle gel-treated EBA ears. $N = 8$ for each group. **f** Representative IHC images of both vehicle and VTI-1002 gel-treated local EBA mouse ears at day 3 stained with neutrophil elastase antibody and 4′,6-diamidino-2-phenylindole (DAPI). White dotted lines indicate the dermal–epidermal junction. Scale bar, 100 μm. Left and right dot plots indicate all individual scores and box plots indicate mean ± standard deviation of relative total cell infiltration score and relative neutrophil infiltrating score, respectively. Both scores are presented relative to the mean scores of vehicle gel-treated EBA ears. $N = 8$ for each group. In the all plots, NS = not significant, *$P < 0.05$, ***$P < 0.001$ (two-sided Student's $t$-test).

functional role for basophils in human PDs. Specifically, GzmB-expressing basophils at the dermis were detected in the majority of human BP patients, though the degree of infiltration varied. Since human basophils are activated through IgE receptor but not through IgG receptors[27,47], we tested if levels of IgE at the lesional or perilesional skin were correlated with infiltrated GzmB-positive basophils. As expected, deposited IgE (detected at the infiltrated cells in most patients[48]) was positively correlated with the number of GzmB-expressing basophils. This result suggests that the variability in GzmB-expressing basophils in human BP is dependent on the variability in IgE levels in human BP skins. In contrast, consistent with former reports indicating that murine basophils are activated through IgG-mediated stimulation[49], our pathogenic IgG transfer adult EBA murine models demonstrated consistent GzmB expression in basophils.

Neutrophil recruitment by mast cells and basophils has been observed in the Kit- or Scf-dependent mast cell deficiency neonatal BP model described above[37] and other disease models[50,51], respectively. While mast cells have previously been reported to upregulate IL-8 production in several cell types (but not keratinocytes) in a tryptase-dependent manner in vitro[52,53], our current study suggests that mast cell-derived GzmB is a key mediator of IL-8 induction in PDs. Conversely, the mechanism underlying neutrophil recruitment by basophils has not been previously elucidated. Our study suggests that basophils may recruit neutrophils through IL-8 production from extracellular GzmB-stimulated keratinocytes.

The current standard of care for PDs, whole body topical or oral corticosteroid administration, is often accompanied by significant and occasionally fatal adverse events[4]. This clinical challenge pertaining to oral corticosteroid use is amplified in PDs compared to other autoimmune diseases since PDs, especially BP, typically occur in the elderly, who are more susceptible to adverse effects of systemic corticosteroid administration[54]. Therefore, combined with the aging population in developed countries, there is an increased impetus to develop safer therapeutic options for PDs. To address this challenge, we validated the therapeutic efficacy of a topical GzmB inhibitor VTI-1002 gel on murine models of PDs. In a previous study, VTI-1002 was demonstrated to potently and specifically inhibit GzmB[24]. In this present study, GzmB was inhibited topically with VTI-1002 gel and not systemically in the murine models as we showed in the current study that GzmB levels in human patients are elevated only at the lesion but not in sera. Topical VTI-1002 penetrates the stratum corneum, is retained in the skin for 24 h with minimal systemic absorption, and does not induce any adverse events or signs of discomfort after 30 days of daily application[24]. In the present study, daily administration of VTI-1002 gel on the PD mice decreased clinical severities, inhibited the loss of hemidesmosomal proteins, and neutrophil infiltration, which is

consistent with findings derived from GzmB−/− mice. In addition, in our current direct immunofluorescence (DIF) analyses, GzmB did not affect IgG and C3 production/deposition at the dermal–epidermal junction. Therefore, VTI-1002 could be a useful therapeutic option in conjunction with current approved medications that can deplete pathogenic IgG production or inhibit IgG recruitment to the dermal–epidermal junction. Importantly, VTI-1002 gel showed therapeutic efficacy in both adult EBA and neonatal BP models, suggesting that VTI-1002 attenuates pathogenicity of GzmB in PDs irrespective of the cell source. Therefore, we expect that application of VTI-1002 gel is a potent therapeutic approach for human PDs.

In summary, GzmB deficiency or inhibition reduces clinical severity and sub-epidermal blistering through the inhibition of neutrophil accumulation and hemidesmosomal protein cleavage in experimental models of PDs. Data in this study support GzmB inhibition as a promising therapeutic approach for the treatment of PDs.

## Methods

**Human samples.** In accordance with Vancouver General Hospital (Vancouver, BC, Canada) and Osaka City University Hospital (Osaka, Japan) research guidelines and policies, we obtained informed patient consent from each participant. All experimental procedures using human samples were approved by the University of British Columbia (UBC) Review Ethics Board and Osaka City University Hospital Review Ethics Board. BP samples including sera, blister fluids, and lesional skins were obtained from patients diagnosed with BP based on clinical, histological, and immunopathological findings.

To purify BP-IgG, BP sera was dialyzed with phosphate-buffered saline (PBS) by 10K MWCO Slide-A-Lyzer dialysis cassettes (Thermo Fisher Scientific, Waltham, MA, USA) and affinity purified with HiTrap protein A columns (GE Healthcare Life Sciences, Marlborough, MA, USA). Purified IgG was concentrated using 100K MWCO Amicon Ultra Centrifuge Filters (Millipore Sigma, St. Louis, MO, USA) followed by dialysis with PBS using 10K MWCO Slide-A-Lyzer dialysis cassettes. Purity of IgG was confirmed with Coomassie staining (Millipore Sigma).

pHEKs were isolated from de-identified human skin samples of healthy adult donors undergoing abdominoplasties. Briefly, skin tissues were processed with 2.4 U/ml of dispase II (Millipore Sigma) at 4 °C overnight to isolate the epidermis from the dermis. The isolated epidermis was trypsinized to obtain keratinocytes.

**Cell culture and GzmB treatment conditions.** pHEKs were cultured in 154 CF medium (Thermo Fisher Scientific) supplemented with 1% Human Keratinocyte Growth Supplement (Thermo Fisher Scientific), 0.07 mM CaCl$_2$ (Thermo Fisher Scientific), and 1% penicillin/streptomycin mixture (Thermo Fisher Scientific) in 5% CO$_2$ at 37 °C. The medium was changed to 154 CF medium supplemented with 1.8 mM CaCl$_2$ 24 h before each experiment. For the GzmB treatment studies, GzmB was pretreated with 1 mM of VTI-1002 or vehicle (dimethyl sulfoxide) in 154 CF medium with 1.8 mM CaCl$_2$ for 30 min at 37 °C. VTI-1002 or vehicle-pretreated GzmB-containing medium was applied to the pHEKs and incubated for 6 h for the hemidesmosomal protein degradation, trypsinization, and cell viability assays, and 16 h for the IL-8 secretion assay. For the neutrophil elastase treatment studies, neutrophil elastase-containing 154 CF medium with 1.8 mM CaCl$_2$ was added to the pHEKs and incubated for 6 h. For the hemidesmosomal protein degradation and IL-8 secretion assays, whole cell lysates and supernatants were collected, respectively. For the trypsinization assay, after the 6-h incubation, the

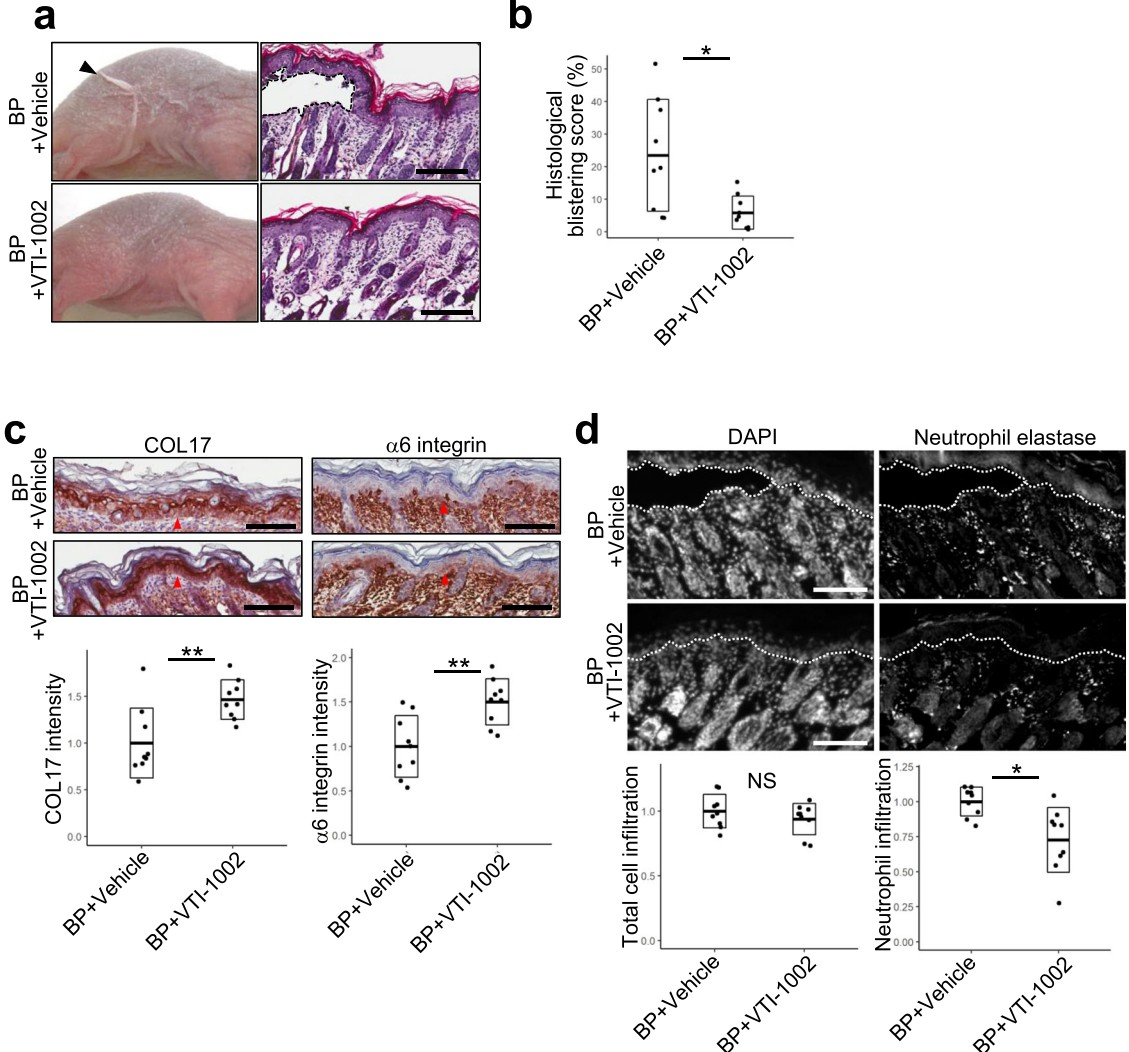

**Fig. 5 Topical VTI-1002 reduces disease severity in a murine bullous pemphigoid (BP) model. a** Representative clinical and H&E staining images of both vehicle and VTI-1002 gel-treated BP mouse at 48 h induced by intraperitoneal BP-IgG injection, labeled as BP+Vehicle and BP+VTI-1002 in the figure, respectively. An arrowhead indicates skin detachment and a dotted line demarcates major dermal–epidermal separation. Scale bar, 100 μm. **b** Histological blistering scores were quantified from H&E staining images. Dot plot indicates all individual scores and box plots indicate mean ± standard deviation. $N = 9$ for each group. **c** Representative type XVII collagen (COL17) and α6 integrin immunohistochemistry (IHC) images of both vehicle and VTI-1002 gel-treated BP mice at 48 h. Red arrowheads indicate positively stained areas at dermal–epidermal junction. Scale bar, 100 μm. Dot plot indicates all individual scores and box plots indicate mean ± standard deviation of the intensity levels of COL17 and α6 integrin, which were quantified from IHC images and presented as relative to the mean score of vehicle gel-treated BP mice. $N = 9$ for each group. **d** Representative IHC images of both vehicle and VTI-1002 gel-treated BP mice at 48 h stained with neutrophil elastase antibody and 4′,6-diamidino-2-phenylindole (DAPI). White dotted lines indicate the dermal–epidermal junction. Scale bar, 100 μm. Left and right dot plots indicate all individual scores and box plots indicate mean ± standard deviation of relative total cell infiltration score and relative neutrophil infiltrating score, respectively. Both scores are presented as relative to the mean scores of vehicle gel-treated BP mice. $N = 9$ for each group. In the all plots, NS = not significant, *$P < 0.05$, **$P < 0.01$ (two-sided Student's $t$-test).

**Table 1 VTI-1002 gel treatment inhibits skin fragility in BP murine model 48 h after the BP-IgG injection. Skin fragility in vehicle or VTI-1002 gel-treated mice was tested for Nikolsky's sign by gently rubbing their flanks three times.**

| Treatment | Skin detachment on either side of flanks | Skin detachment on both sides of flanks |
|---|---|---|
| Vehicle gel application | 9/9 | 7/9 |
| VTI-1002 gel application | 3/9 | 1/9 |

cells were washed three times with PBS and then incubated with 0.25% trypsin (Thermo Fisher Scientific) for 7 min at 37 °C. After three PBS washes, the number of attached cells were counted. For the cell viability assay, after GzmB or neutrophil elastase treatment, the cells were incubated with a final concentration of 0.5 mg/ml MTT for 3 h at 37 °C. MTT was then removed and the cells were lysed in dimethyl sulfoxide/ethanol (1:1) solvent. Absorbance at 565 nm was measured by Infinite

M1000 Pro plate reader (TECAN, Männedorf, Switzerland). Percent cell viability was calculated with a formula: ((absorbance from sample well)−(absorbance from blank well))/((absorbance from control well)−(absorbance from blank well)) × 100. Passages up to 7 were used for all experiments.

Primary mouse epidermal keratinocytes were isolated from humanized COL17 mouse neonatal skin samples. Briefly, skin tissues were processed with 2.4 U/ml of

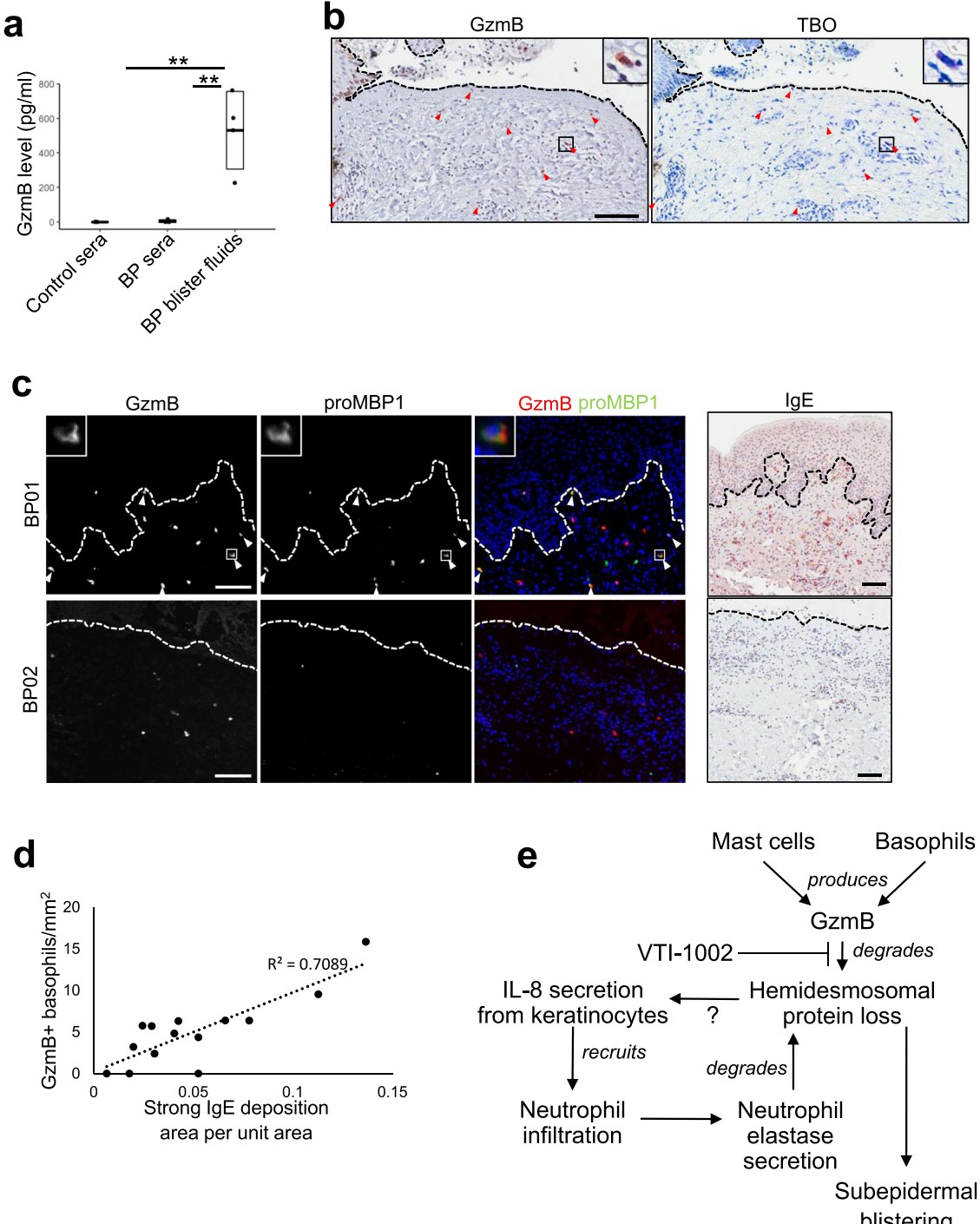

**Fig. 6 GzmB levels are elevated in primary human bullous pemphigoid (BP) blister fluids and lesional skin. a** Dot plot indicates all individual concentrations and box plots indicate mean ± standard deviation of human granzyme B (GzmB) levels in human control sera ($N = 3$), BP patient sera ($N = 4$), and BP patient blister fluids ($N = 4$), as quantified by enzyme-linked immunosorbent assays (ELISA). **$P < 0.01$ (one-way ANOVA followed by Tukey's multiple pairwise-comparisons). **b** Representative sequential GzmB immunohistochemistry (IHC) and toluidine blue O (TBO) staining image of human BP lesional skin. Red arrowheads indicate cells stained with both GzmB and TBO and dotted lines demarcate a histological subepidermal blister. Scale bar, 100 μm. **c** Representative double staining images of human BP skin with antibodies against GzmB and human basophil marker pro-major basic protein 1 (proMBP1) and representative IHC images of human BP skin stained with anti-IgE antibody. Third column shows overlay of two stains. White arrowheads indicate cells stained with both GzmB and proMBP1. Dotted lines demarcate a histological subepidermal blister. Scale bar, 100 μm. Images in **b** and **c** are representative of two independent experiments. **d** Scatter plot indicates scores of strong IgE deposition area per unit area in relation to the number of GzmB-expressing basophils/mm$^2$. Dotted line indicates regression line. **e** Proposed model for the role of GzmB in pemphigoid disease (PD) pathology. Mast cells and/or basophils produce GzmB to degrade hemidesmosomal proteins. Loss of hemidesmosomal proteins induces not only dermal–epidermal attachment loss resulting in subepidermal blistering but potentially IL-8 secretion from keratinocytes also. IL-8 recruits neutrophils into the upper dermis to induce additional hemidesmosomal protein degradation, which leads to further subepidermal blistering.

dispase II (Millipore Sigma) at 4 °C overnight to isolate the epidermis from the dermis. The isolated epidermis was processed with TrypLE Select (Thermo Fisher Scientific) to obtain keratinocytes. Primary mouse epidermal keratinocytes were cultured in CnT-07 medium (CELLnTEC, Bern, Switzerland) supplemented with 1% penicillin/streptomycin mixture (Thermo Fisher Scientific) in 5% $CO_2$ at 37 °C. The medium was changed to CnT-07 medium supplemented with 1.8 mM $CaCl_2$ 24 h before the MIP-2 secretion assays. For the MIP-2 secretion assays, VTI-1002 or vehicle-pretreated GzmB-containing medium was added to the primary mouse epidermal keratinocytes and incubated for 16 h before supernatants were collected.

NK cell line YT-INDY cells were cultured in RPMI-1640 (Millipore Sigma) supplemented with 10% fetal bovine serum (Thermo Fisher Scientific), 1% MEM non-essential amino acid solution (Thermo Fisher Scientific), and 1% penicillin/ streptomycin mixture in 5% $CO_2$ at 37 °C.

**GzmB purification and VTI-1002**. GzmB was purified from the isolated granules of YT-INDY using a procedure adapted from a former study[55]. Briefly, the cell pellet was suspended in PBS containing EGTA (5 mM) and magnesium chloride (1 mM) and subjected to three freeze–thaw cycles alternating between −80 °C and 37 °C. The solution was then centrifuged and the supernatant was collected. Triton X-100 was added to a final concentration of 1% and incubated for 10 min at room temperature, followed by centrifugation. The supernatant was collected and filtered through a 0.45-μm filter and applied to a HiTrap Heparin column (GE Healthcare Life Sciences). The column was washed and a stepwise elution was performed by eluting first with elution buffer A (20 mM HEPES, 250 mM NaCl pH 6.0), followed by elution buffer B (20 mM HEPES, 500 mM NaCl pH 6.0), and finally elution buffer C (20 mM HEPES, 1000 mM NaCl pH 6.0). GzmB-positive fractions and these activities were identified using the IEPD-AMC substrate (GzmB substrate) (Enzo Life Sciences, Inc., Farmingdale, NY, USA). Briefly, each fraction was mixed with 50 mM Tris pH 7.0 and incubated with 50 μM of IEPD-AMC for 30 min at room temperature in a black 96-well flat bottom plate (Millipore Sigma) and the fluorescence was read with an Infinite M1000 Pro plate reader. A kinetic study was undertaken and the fluorescence read every minute for 15 min (Supplementary Fig. 2a). Recombinant human GzmB (Emerald BioSystems, Bainbridge Island, WA, USA) was used as a standard.

Purified GzmB was analyzed for protease activities of neutrophil elastase, MMP-2/-9, and plasmin, which were not detected (Supplementary Fig. 2b–d). Briefly, 50 nM of purified GzmB in 50 mM Tris pH 7.0 was incubated at room temperature with 50 μM of neutrophil elastase substrates (Biovision, Milpitas, CA, USA), MMP-2/MMP-9 substrates (Millipore Sigma), or plasmin substrates (Biovision) and absorbance or fluorescent intensity was measured every minute for 15 min or for 60 min as indicated in the figures with Infinite M1000 Pro plate reader. Neutrophil elastase (Athens Research and Technology Inc., Athens, GA, USA), MMP-9 catalytic domain (Anaspec Inc., Fremont, CA, USA), and plasmin (Millipore Sigma) were used as standards.

GzmB-specific small molecule inhibitor VTI-1002 was provided from viDa therapeutics, Inc (Vancouver, BC, Canada). Characterization of VTI-1002 was described elsewhere[24]. VTI-1002 was dissolved to 100 mM in dimethyl sulfoxide for in vitro cell experiments. For topical application studies, VTI-1002 was formulated to 3.6 mg/ml in a vehicle gel consisting of carbopol, propylene glycol, methyl paraben, and propyl paraben in acetate buffer, as previously described[24].

**Cleavage assay for recombinant COL17**. Recombinant GzmB was pretreated with 1 mM of VTI-1002 or vehicle (dimethyl sulfoxide) in 50 mM Tris pH 7.5 for 60 min at 37 °C; 500 ng of recombinant COL17 was added to VTI-1002 or vehicle-pretreated GzmB-containing buffer and incubated for 2 h before immunoblotting analyses.

**Laboratory mice**. C57Bl/6 (WT) and GzmB−/− mice with C57Bl/6 background were obtained from Jackson Laboratory (Bar Harbor, ME, USA). GzmB−/− mice were maintained as homozygous colonies or by continuous backcross to C57Bl/6 mice. Humanized COL17 mice (mCol17−/−, hCOL17Tg/Tg) were generated as described elsewhere[25] and were maintained as homozygous colonies. WT, GzmB−/−, and humanized COL17 mice were bred and housed in a 12 h light/dark cycle with controlled room temperature at 20–26 °C and 40–60% relative humidity at the Genetic Engineered Models facility, St. Paul's Hospital, UBC or at ICORD, UBC. All procedures were performed in accordance with the guidelines for animal experimentation approved by the Animal Experimentation Committee of UBC. Double transgenic Cre/iDTR mice were generated by crossing Mcpt5-Cre mice (C57Bl/6 background) with the iDTR line as described elsewhere[56,57]. This strain was bred and housed at the animal facilities of the Research Center Borstel.

**Animal models**. Systemic antibody-transfer model of EBA was established as described elsewhere (reviewed in ref. [20]). Briefly, 3.5 mg of rabbit anti-mouse COL7 IgG or control normal rabbit IgG (Lampire Biological Laboratories, Inc., Pipersville, PA, USA) was intraperitoneally injected once every second day over 10 days into WT or GzmB−/− mice at 7–10 weeks old. Visible erythema, eruption, and ulcer were defined as lesions. Estimated whole lesional area was divided by the estimated full body surface area ($9.82 \times$ body weight (g) $\wedge$ 0.667)[58] in order to

quantify the percentage of lesional area, defined as the affected body surface area. The mice were euthanized at day 12 and the samples were collected for analyses.

Local antibody-transfer model of EBA was induced using a procedure adapted from a protocol described elsewhere (reviewed in ref. [20]). Briefly, 1 mg of rabbit anti-mouse COL7 IgG was subcutaneously injected into the ear tip of WT mouse at 7–10 weeks old at day 0 and day 1. Fifty microliters of 3.6 mg/ml VTI-1002 gel or vehicle gel was topically applied onto each ear every day from day −1. Visible erythema and crusts were defined as lesions. Percentage of affected ear surface was quantified at day 3. Ear thickness was measured with a digital caliper. For each mouse, at least three different sites of the ear were measured and the average thickness was calculated. The mice were euthanized at day 3 and the samples were collected for analyses.

The BP-IgG transfer model of BP was established as described elsewhere[25]. Briefly, 1-day old neonatal humanized COL17 mice received intraperitoneal injection of 1 mg/g BP-IgG; 100 μl of the VTI-1002 gel or vehicle gel was topically applied onto the entire body every day from day −1. Forty-eight hours after the injection, the mice were gently rubbed three times to test for Nikolsky's sign. The mice were then euthanized and samples were collected for analyses.

Mast cell-deficient mice were established with four peritoneal DT injections in 6- to 9-week-old Cre/iDTR mice as indicated elsewhere[56]. Paraffin-embedded mast cell-deficient mouse samples obtained in a previous experiment[56] were utilized for the current study to reduce the number of animals used.

Sample sizes for all animal studies were determined by statistical power calculations prior to the experiments based on standard deviations estimated from our former[24] or preliminary results.

**Antibodies**. Rabbit polyclonal antibodies against GzmB (ab4059) and neutrophil elastase (ab68672) and rabbit monoclonal antibodies against β4 integrin (EPR17517, ab182120), the 1,300–1,400 amino acid region of COL17 (EPR18614, ab184996), and α6 integrin (EPR18124, ab181551) were purchased from abcam (Cambridge, MA, USA). Rat monoclonal antibody against mMCP-8 (TUG8) and mouse monoclonal antibody against human ProMBP1 in basophils (J175-7D4) were purchased from BioLegend (San Diego, CA, USA). Rabbit polyclonal antibody against NC16A domain of COL17 was a generous gift from Dr. Claus-Werner Franzke (University of Freiburg). Mouse monoclonal antibody against β-tubulin (AA2) was purchased from Millipore Sigma. Rabbit monoclonal antibody against glyceraldehyde 3-phosphate dehydrogenase (GAPDH) (14C10, #2118) was purchased from Cell Signaling Technology (Danvers, MA, USA). Fluorescein-conjugated goat polyclonal antibody against mouse complement C3 (55500) was purchased from MP Biomedicals (Solon, OH, USA). Rabbit anti-human IgE heavy chain (PA5-16396) was purchased from Thermo Fisher Scientific. Alexa Fluor 488- and Alexa Fluor 594-conjugated donkey anti-rabbit-IgG and donkey anti-rat-IgG antibodies were purchased from Thermo Fisher Scientific (A21206, A21207, A21208, A21209). Biotin-conjugated goat anti-rabbit-IgG (BA-1000) and horse anti-mouse-IgG (BA-2000) antibodies were purchased from Vector Laboratories (Burlingame, CA, USA). HRP-conjugated rabbit anti-mouse IgG (SC-358914) and goat anti-rabbit IgG (SC-2054) antibodies were purchased from Santa Cruz Biotechnology (Dallas, TX, USA).

**Histology, TBO staining, and IHC**. Five-micrometer sections of paraffin-embedded skin tissue blocks were deparaffinized and rehydrated for histological, TBO staining, and immunohistochemical analyses. For histological analyses, H&E staining was performed. Histological blister score was calculated using the formula: ((combined total length of all blistered regions)/(combined total length of all dermal–epidermal junction examined)) × 100, adapted from a formula described elsewhere[59]. For TBO staining, the sections were stained with 0.1% TBO solution at pH 2.0–2.5. After imaging, the sections were processed for IHC with GzmB antibody. For IHC, the sections were processed for antigen retrieval by incubating with trypsin (Carezyme I: Trypsin Kit, Biocare Medical LLC, Pacheco, CA, USA) at room temperature for 30 min for double staining of GzmB and human basophils or by boiling in citrate buffer (Invitrogen, Carlsbad, CA, USA) for others, followed by blocking with 5% serum for 1 h at room temperature. The sections were incubated with primary antibodies with 1:100 dilution except 1:50 dilution for mMCP-8 antibody (TUG8) and 1:200 dilution for α6 integrin antibody (EPR18124) at 4 °C overnight. For chromogenic detection in IHC, the sections were incubated with diluted biotin-conjugated secondary antibodies at room temperature for 1 h and the sections were processed with Vectastain ABC kit (Vector Laboratories, Burlingame, CA) and counter-staining in methyl green. All H&E, TBO staining, and chromogenically stained immunohistochemical slides were imaged with the Aperio CS2 slide scanner (Leica Biosystems, Buffalo Grove, IL, USA). The images were prepared using the Aperio ImageScope Viewer (Leica Biosystems). The intensity levels of hemidesmosomal proteins in epidermis was quantified using a formula: (total intensity levels of strong positive in epidermis)/(total epidermis area examined). The levels of IgE deposition in dermis were quantified using the formula: (total area of strong positive in dermis)/(total dermis area examined), both of which were measured using the Aperio ImageScope Viewer. The threshold for 'strong positive' was determined in order to exclude weak positive signals in the dermis; same threshold was used for all samples. For fluorescence detection in IHC, the sections were incubated with Alexa Fluor 488- and/or 594-conjugated secondary antibodies (1:500 dilution) followed by 5-min incubation with 4′,6-diamidino-2-phenylindole (DAPI) (Millipore Sigma). For fluorescence detection of J175-

7D4, the sections were incubated with HRP-conjugated secondary antibodies followed by TSA amplification with TSA Plus Fluorescein Evaluation kit (NEL741E001KT, PerkinElmer Inc, Waltham, MA, USA) and then incubated with DAPI. The fluorescently stained IHC slides were imaged with the EVOS FL Imaging System (Thermo Fisher Scientific). The images were prepared and analyzed using ImageJ (National Institutes of Health, Bethesda, MD, USA). To quantify infiltrating cells and neutrophils at the upper dermis, the neutrophils and whole cells in the upper dermis were selected using the threshold function in ImageJ after selection of the upper dermis area. Areas of hair follicles and vesicles were carefully excluded. Total cell infiltration score ((total DAPI-positive area in upper dermis)/(total area of upper dermis) × 100) and neutrophil infiltrating score ((total neutrophil elastase-positive area in upper dermis)/(total DAPI-positive area in upper dermis) × 100) were calculated from the images.

**DIF analyses**. Optimal cutting temperature compound (Thermo Fisher Scientific) embedded samples were cut into 10-µm sections for DIF analyses. For rabbit IgG staining, the sections were incubated with Alexa Fluor 488-conjugated antibody against rabbit IgG (1:500) for 1 h at room temperature. For mouse C3 staining, the sections were first fixed with 3.7% formaldehyde for 10 min followed by incubation with fluorescein-conjugated goat antibody against mouse complement C3 (1:100) for 1 h at room temperature. The slides were imaged with EVOS FL Imaging System and three representative images were taken from each ear sample. The images were prepared and analyzed using ImageJ. The linear staining pattern at the dermal–epidermal junction were selected by the wand tool to quantify the mean intensities at the region. The average of the mean intensities from each sample was calculated.

**SDS-PAGE and immunoblot analyses**. Liquid-nitrogen-frozen ears were ground and re-suspended in Laemmli buffer. Whole cell extracts were prepared in CelLytic MT Cell Lysis Reagent (Millipore Sigma) and diluted in Laemmli buffer. Protein concentrations were determined with the Pierce BCA Protein Assay Kit (Thermo Fisher Scientific), according to the manufacturer's protocol. A total of 5% of dithiothreitol was added to the extracts and then heated at 95 °C for 5 min. Reduced extracts were resolved by SDS-PAGE and transferred onto nitrocellulose membrane for immunoblot analysis. Transferred membranes were blocked with 5% non-fat milk in TBS supplemented with 0.05% Tween-20 and then probed with primary antibodies (1:1000 dilution) except COL17 NC16a antibody (1:2000 dilution) in blocking buffer at 4 °C overnight. Membranes were then incubated with 1:3000 dilution of secondary antibodies in blocking buffer for 1 h. Immunoblots were visualized using an Odyssey Blot Imager (LI-COR Biosciences, Lincoln, NE, USA) and quantified using ImageJ. Uncropped and unprocessed scans of the blots are included in the Source Data file.

**Protease activity assays**. Liquid-nitrogen-frozen mouse lesional skins were ground and re-suspended in 50 mM Tris pH 7.5 for protease activity assays. Homogenized skin specimens were incubated with 50 µM of neutrophil elastase substrates, MMP-1/MMP-9 substrates (Bachem Americas, Inc., CA, USA), or plasmin substrates at room temperature. Its absorbance or fluorescent intensity was measured immediately and 30 min after with Infinite M1000 Pro plate reader and Δ optical density (OD) or Δ relative fluorescent unit (RFU) was calculated. ΔRFU and ΔOD were normalized to the average of them from WT mice with EBA.

**Enzyme-linked immunosorbent assays (ELISA)**. Liquid-nitrogen-frozen mouse lesional skins were ground and re-suspended in RIPA buffer for MIP-2 ELISA. Protein concentrations were determined with the Pierce BCA Protein Assay Kit. Concentrations of MIP-2, IL-8, and human GzmB were determined with the MIP-2 (ELM-MIP2-1, Ray Biotech, Peachtree Corners, GA, USA), IL-8 (ab214030, Abcam), and human Granzyme B (BMS2027, Thermo Fisher Scientific) ELISA kits, respectively, according to the manufacturers' protocols.

**Statistics**. All experiments were performed at least three times and analyzed using R (R Foundation for Statistical Computing, Vienna, Austria). Statistical significance was determined by Student's $t$-test or ANOVA followed by Tukey's multiple pairwise-comparisons and indicated by a bar and asterisk above each data set. A value of $P < 0.05$ was considered significant. Where no significance was detected, no bar is included in the figures.

**Reporting summary**. Further information on research design is available in the Nature Research Reporting Summary linked to this article.

## Data availability

The data supporting the finding of this study are available within the article and from the corresponding author on reasonable request. Source data are provided with this paper.

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

## Acknowledgements

We express our thanks to Dr. Karen Jung and Aoi Hiroyasu from the Granville laboratory, ICORD, UBC, and Tatjana Bozin from the Centre for Heart Lung Innovation, St. Paul's Hospital, UBC, for their assistance with manuscript editing, experiments, and animal husbandry, respectively. This research was supported by funding from the Canadian Institutes for Health Research (CIHR) (D.J.G.) and the Michael Smith Foundation for Health Research (MSFHR) (D.J.G.). S.H. and C.T.T. are supported by CIHR Post-doctoral Fellowships. M.R.Z. is supported by a CIHR Post-doctoral Fellowship and a MSFHR Post-doctoral Fellowship.

## Author contributions

S.H. and D.J.G. designed the research studies. S.H., M.R.Z., H.Z., M.A.P., and C.T.T. conducted the experiments. S.H., R.J.L., and D.J.G. interpreted the data. A.K., C.T., W.N., A.B., P.A.L., N.V.L., N.J.C., F.P., R.I.C., H.S., D.T., and R.J.L. provided reagents, experimental assistance, input, and insights into manuscript preparation. S.H. and D.J.G. wrote and prepared the manuscript.

## Competing interests

D.J.G. is a Co-Founder, Chief Scientific Officer, and Consultant of viDA Therapeutics, Inc., which developed VTI-1002, a granzyme B inhibitor used in this manuscript. The remaining authors have no competing interests to declare.
