## [Peer Review File · Nature Communications]

Reviewers' comments:

Reviewer #1 (Pemphigus and models)(Remarks to the Author):

In this manuscript, the authors found that GzmB is elevated in human PD blister fluids and lesional skin. Mice lacking GzmB activity (knockout or pharmacologic topical inhibition) develop significantly reduced subepidermal blistering in animal models of EBA and BP. While these observations implicate GzmB in the development of these PDs, the authors fail to demonstrate what roles other proteases, such as neutrophil elastase and MMP-9 critical for these PDs in mice play in their experimental setting and relative contributions of GzmB and NE and MMP-9 in the disease development. Another concern is the clinical relevance of the presented findings.

Major issues:

1. Neutrophils, MMP-9 and plasmin were shown to be involved in these PDs in mice. The authors should quantify levels of these proteases in their diseased mice, antibody-injected GzmB knockouts and GzmB inhibitor treated mice. Additionally, the relative contributions of these proteases to overall disease activity are both interesting and of clinical significance.
2. Will disease severity increase in antibody injected GzmB mice if the time course is extended?
3. In EBA models, GzmB are identified in mast cells and basophils. Since mast cells are dispensable in EBA mouse models, it is important to know if basophils and GzmB in basophils are critical for EBA in mice. A more fundamental question is whether the findings related to basophils in EBA mouse models are clinically relevant given the fact that human basophils cannot be activated through IgG and IgG receptor interaction.
4. The authors show reduced level of full-length col 17 in GzmB treated keratinocytes. To directly demonstrate and confirm that GzmB cleaves Col 17, the author should use cell-free system and show degradation of purified col 17 by purified GzmB.

Minor points

1. Fig 1D, higher magnification images should be included to better show MCs in the skin.
2. What is the purity of the GzmB preparation? The authors should determine whether the purified GzmB are contaminated with other proteases?
3. Fig 3D. IL-8 is released in pHEKs incubated with GzmB. To be consistent and to rule out that mouse and human keratinocytes could respond differently to GzmB, the authors should confirm this finding by incubating GzmB with mouse keratinocytes.

Reviewer #2 (Granzymes)(Remarks to the Author):

This new study from Granville's lab analyses the contribution of the serineprotease granzyme B to the pathology of skin blistering diseases also known as pemphigoid diseases (PDs). The authors use both genetically deficient gzmB mice (gzmB KO) as well as a topic gzmB inhibitor to show that gzmB significantly contributes to PDs by affecting the dermal-epidermal junction. GzmB deficiency or inhibition reduced the degradation of proteins involved in the maintenance of dermal-epidermal junction, specifically collagen XVII and $\alpha 6$ -integrin, correlating with a reduction in both neutrophil infiltration and MCP1 expression in skin. In vitro, gzmB directly cleaves collagen XVII but not $\alpha 6$ -integrin, suggesting that both direct and indirect effects are involved in gzmB-mediated cleavage of dermal-epidermal junctions. In addition, gzmB is able to induce IL8 release from keratinocytes in vitro, which could enhance skin inflammation and blistering severity. GzmB is mainly expressed in skin-associated mast cells and basophils (depending on the model) of mice suffering from blistering, and these cells are suggested to be the main source of extracellular gzmB responsible for skin pathology. This is an elegant study and the results are highly relevant since they suppose a significant advance on the mechanism involved in blistering and provides a new potential target to treat this disease in a more selective and safer way than the current immunosuppressors commonly used to treat this affection. The translational relevance of the findings are supported by

showing that gzmB is enhanced in human PD blister fluids and affected tissues. The experiments are well controlled and the results are in accordance with the main conclusions, supporting an important role for extracellular gzmB in blistering.

This reviewer only has minor concerns regarding the interpretation of some of the results and the accuracy of the final scheme summarizing the contribution of extracellular gzmB to blistering. In addition, I would like to suggest some additional controls to strength the main conclusions of the paper regarding the contribution of gzmB-mediated extracellular matrix degradation to blistering severity.

1- Although it is true that gzmB deficiency has a lower impact than other granule components like perforin, in the control of tumor development and infection, based on the previous published data, I would suggest rephrasing the sentence (line 77 and 309) "... gzmB^{-/-} mice do not exhibit susceptibility to viral infections or tumorigenesis..." Authors cite two papers to support this claim. However, other studies suggest that deletion of gzmB might affect tumor development (Pardo et al, EJI, 2002, Revell et al JI 2003) and the control of some infections like ectromelia, gammaherpes, LCMV or Brucella. I agree that inhibition of gzmB (specially locally) might have minimal effects regarding cancer development and infection, since in most cases a delay in control is observed in gzmB Ko mice. However, based on all these results I would suggest changing these sentences.

2- As indicated in the discussion, other proteases derived from mast cells might contribute to the degradation of ECM proteins and the loss of dermal-epidermal junctions. From the in vivo data, it seems that gzmB is required for loss of dermal-epidermal junctions and degradation of ECM proteins like collagen XVII and $\alpha 6$ -integrin. However, meanwhile gzmB directly cleaves CXVII, the same effect is not observed for integrin. Thus, it is plausible that other proteases activated by gzmB and secreted extracellularly contribute to degradation of $\alpha 6$ -integrin and PDs. It has been recently shown that mast cells granules contain and secrete active caspase-3, by a gzmB-restricted mechanism (Zorn et al EJI 2013; Garcia-Faroldi et al JI 2013). Since this protease is known to degrade several substrates including some integrins, it would be nice to discuss a potential role for this mechanism in gzmB-mediated $\alpha 6$ -integrin degradation observed in vivo.

3- Have the authors analysed the expression of MIP-2 in vitro in gzmB-stimulated keratinocytes? On the same way, is IL8 reduced in gzmB KO mice in vivo in the models of PDs?

4- It would be good to show the viability of keratinocytes exposed to gzmB in figure 2D. The in vivo data nicely show that gzmB deficiency does not affect the number of dead cells in vivo and showing it also in vitro would nicely round up the results.

5- Regarding the scheme in figure 6D, the connection between protein degradation and IL8 secretion by keratinocytes is not proven and thus it should be indicated with a question mark.

Reviewer #1 (Pemphigus and models) (Remarks to the Author):

Major issues:

1. Neutrophils, MMP-9 and plasmin were shown to be involved in these PDs in mice. The authors should quantify levels of these proteases in their diseased mice, antibody-injected GzmB knockouts and GzmB inhibitor treated mice. Additionally, the relative contributions of these proteases to overall disease activity are both interesting and of clinical significance.

REPLY: We agree with the Reviewer's comment and pursued new experiments investigating neutrophil elastase, MMP-9, and plasmin protease activity in WT and GzmB^{-/-} mice with EBA. In **newly-generated Figure 3E**, we show that neutrophil elastase activity is down-regulated by approximately 50% in GzmB^{-/-} mice with EBA compared to WT mice with EBA. In contrast, MMP-9 and plasmin activity were not significantly different between WT and GzmB^{-/-} mice with EBA (**newly-generated Figure 3E**). This new data is in line with our previous findings that neutrophil infiltration (Figure 3B) and neutrophil chemoattractant MIP-2 (Figure 3C) are both reduced in GzmB^{-/-} mice with EBA compared to WT mice with EBA. Furthermore, we examined the pathological significance of neutrophil elastase and show for the first time that neutrophil elastase degrades $\alpha 6$ integrin without affecting the cell viability in **newly-generated supplementary Figures 1E and 1F**. Collectively, we demonstrate that GzmB contributes to pemphigoid disease severity through both direct proteolytic degradation and neutrophil elastase-dependent degradation of hemidesmosomal proteins but not through MMP-9 nor plasmin-dependent mechanisms.

2. Will disease severity increase in antibody injected GzmB mice if the time course is extended?

REPLY: We anticipate that disease severity will increase if the time course and IgG injections are extended. However, in accordance to animal welfare and ethics guidelines, we are not permitted to generate a more severe disease phenotype.

3. In EBA models, GzmB are identified in mast cells and basophils. Since mast cells are dispensable in EBA mouse models, it is important to know if basophils and GzmB in basophils are critical for EBA in mice. A more fundamental question is whether the findings related to basophils in EBA mouse models are clinically relevant given the fact that human basophils cannot be activated through IgG and IgG receptor interaction.

REPLY: To address the Reviewer's comment, we collaborated with newly added coauthor, Dr. Frank Peterson (Borstel, Germany), and confirmed that GzmB is expressed in basophils in mast cell-deficient mice with EBA (**newly-generated Supplementary Figure 1B**). We suspect that GzmB in basophils compensates for GzmB in mast cells and therefore exhibits a critical pathological role. In addition, we have expanded our analysis of BP patients from 3 patients to 14 patients (**revised Figure 6C**). As a result, we found that BP patients express varied numbers of GzmB-expressing basophils which correlate to IgE deposition at the dermis (**revised Figures 6C and D**). We also added the following to the Discussion (page 16, line 2): "the current

study proposes a functional role for basophils in human PDs. Specifically, GzmB-expressing basophils at the dermis were detected in the majority of human BP patients, though the degree of infiltration varied. Since human basophils are activated through IgE receptor but not through IgG receptors (27, 47), we tested if levels of IgE at the lesional or perilesional skin were correlated with infiltrated GzmB-positive basophils. As expected, deposited IgE (detected at the infiltrated cells in most patients previously described (48)) was positively correlated with the number of GzmB-expressing basophils. This result suggests that the variability in GzmB-expression basophils in human BP is dependent on the variability in IgE levels in human BP skins. In contrast, consistent with former reports indicating that murine basophils are activated through IgG-mediated stimulation (49), our pathogenic IgG transferred adult EBA murine models demonstrated consistent GzmB expression in basophils."

We found that the Reviewer's comment regarding the specific function of GzmB in basophils is scientifically intriguing. However, we think that establishing a PD mouse model with GzmB deficiency in basophils is beyond the scope of the current study as the main goals of our study are to define the function and clinical relevance of GzmB in PDs, and to assess GzmB inhibition as a therapeutic mechanism in PDs, irrespective of the cell source(s) of GzmB. To clarify this, we have added the following to our Discussion (page 17, line 19): "Importantly, VTI-1002 gel showed therapeutic efficacy in both adult EBA and neonatal BP models, suggesting that VTI-1002 attenuates pathogenicity of GzmB in PDs irrespective of the cell source. Therefore, we expect that application of VTI-1002 gel is a potent therapeutic approach for human PDs."

4. The authors show reduced level of full-length col 17 in GzmB treated keratinocytes. To directly demonstrate and confirm that GzmB cleaves Col 17, the author should use cell-free system and show degradation of purified col 17 by purified GzmB.

REPLY: We agree with the Reviewer's comment and have now provided new data demonstrating recombinant GzmB cleavage of recombinant COL17 in a cell-free system in **newly-generated Supplementary Figure 1C**.

Minor points:

1. Fig 1D, higher magnification images should be included to better show MCs in the skin.

REPLY: We have now added high power images to **Figure 1D, Supplementary Figure 1B, and Supplementary Figure 2B**.

2. What is the purity of the GzmB preparation? The authors should determine whether the purified GzmB are contaminated with other proteases?

REPLY: To address this comment, we tested the activity of GzmB, neutrophil elastase, MMP-9, and plasmin in our purified GzmB using specific substrates. In **newly-generated Supplementary Figures 3A to 3D**, we show that our purified GzmB is free of contaminating proteases.

3. Fig 3D. IL-8 is released in pHEKs incubated with GzmB. To be consistent and to rule out that mouse and human keratinocytes could respond differently to GzmB, the authors should confirm this finding by incubating GzmB with mouse keratinocytes.

REPLY: To address this comment, we now provide data showing GzmB induction of MIP-2 in primary mouse keratinocytes in **newly-generated Figure 3D**. The original data showing GzmB induction of IL-8 in pHEKs has been moved to Supplementary Figure 1G.

Reviewer #2 (Granzymes) (Remarks to the Author):

From Reviewer #2: "In addition, I would like to suggest some additional controls to strength the main conclusions of the paper regarding the contribution of gzmB-mediated extracellular matrix degradation to blistering severity."

1- Although it is true that gzmB deficiency has a lower impact than other granule components like perforin, in the control of tumor development and infection, based on the previous published data, I would suggest rephrasing the sentence (line 77 and 309) "... gzmB^{-/-} mice do not exhibit susceptibility to viral infections or tumorigenesis..." Authors cite two papers to support this claim. However, other studies suggest that deletion of gzmB might affect tumor development (Pardo et al, EJI, 2002, Revell et al JI 2003) and the control of some infections like ectromelia, gammaherpes, LCMV or Brucella. I agree that inhibition of gzmB (specially locally) might have minimal effects regarding cancer development and infection, since in most cases a delay in control is observed in gzmB Ko mice. However, based on all these results I would suggest changing these sentences.

REPLY: We agree with the Reviewer that our sentence was inaccurate and have now removed the description. However, based on our previous work showing that topical VTI-1002 does not induce any adverse events or signs of discomfort after 30 days of daily application (Shen et al, 2018), we believe that local inhibition with VTI-1002 is relatively safe. This description can be found in the Discussion on page 17, line 11.

2- As indicated in the discussion, other proteases derived from mast cells might contribute to the degradation of ECM proteins and the loss of dermal-epidermal junctions. From the in vivo data, it seems that gzmB is required for loss of dermal-epidermal junctions and degradation of ECM proteins like collagen XVII and $\alpha 6$ -integrin. However, meanwhile gzmB directly cleaves CXVII, the same effect is not observed for integrin. Thus, it is plausible that other proteases activated by gzmB and secreted extracellularly contribute to degradation of $\alpha 6$ -integrin and PDs. It has been recently shown that mast cells granules contain and secrete active caspase-3, by a gzmB-restricted mechanism (Zorn et al EJI 2013; Garcia-Faroldi et al JI 2013). Since this protease is known to degrade several substrates including some integrins, it would be nice to discuss a potential role for this mechanism in gzmB-mediated $\alpha 6$ -integrin degradation observed in vivo.

REPLY: To address the Reviewer's comment, we have now added the following to the Discussion on page 14, line 15: "While this is the first report describing GzmB-mediated induction of neutrophil elastase activity to enhance hemidesmosomal protein loss in PDs, this does not exclude potential interactions between GzmB and other proteases in PDs. Indeed, a recent work revealed that GzmB activates caspase 3 in secretory lysosomes of mast cell, a mechanism which

possibly contributes to enhanced caspase 3 dependent proteolytic cleavage in the extracellular space (39). Since recent studies combining genomics, proteomics, and bioinformatics are beginning to elucidate the direct and indirect mutual influence of proteases (40), our understanding of the complex interactions between GzmB and other proteases in PDs will be further refined."

3- Have the authors analysed the expression of MIP-2 in vitro in gzmB-stimulated keratinocytes? On the same way, is IL8 reduced in gzmB KO mice in vivo in the models of PDs?

REPLY: This was also suggested in Reviewer #1's minor point #3. To address this comment, we now provide data showing GzmB induction of MIP-2 in primary mouse keratinocytes in **newly-generated Figure 3D**. The original data showing GzmB induction of IL-8 in pHEKs has been moved to Supplementary Figure 1G. Moreover, we have demonstrated in Figure 3C that IL-8 mouse homologue, MIP-2, is reduced in GzmB^{-/-} mice with EBA compared to WT mice with EBA.

4- It would be good to show the viability of keratinocytes exposed to gzmB in figure 2D. The in vivo data nicely show that gzmB deficiency does not affect the number of dead cells in vivo and showing it also in vitro would nicely round up the results.

REPLY: To address the Reviewer's comment, we have now confirmed in an MTT assay that GzmB stimulation does not affect keratinocyte viability (**newly-generated Supplementary Figure 1D**).

5- Regarding the scheme in figure 6D, the connection between protein degradation and IL8 secretion by keratinocytes is not proven and thus it should be indicated with a question mark.

REPLY: We agree with the Reviewer's comment and have now added a question mark to denote the connection in question in new Figure 6E (original Figure 6D).

REVIEWERS' COMMENTS

Reviewer #1 (Remarks to the Author):

None. The authors have adequately addressed all points raised by this reviewer.

Reviewer #2 (Remarks to the Author):

The authors have properly addressed all my concerns. I would like to thank the detailed revision of all my comments.